# A Multimodal Biomarker Predicts Dissemination of Bronchial Carcinoid

**DOI:** 10.3390/cancers14133234

**Published:** 2022-06-30

**Authors:** Ellen M. B. P. Reuling, Dwayne D. Naves, Pim C. Kortman, Mark A. M. Broeckaert, Peter W. Plaisier, Chris Dickhoff, Johannes M. A. Daniels, Teodora Radonic

**Affiliations:** 1Department of Surgery, Amsterdam University Medical Center, VUMC, University Amsterdam, 1081 HV Amsterdam, The Netherlands; e.reuling@amsterdamumc.nl (E.M.B.P.R.); c.dickhoff@amsterdamumc.nl (C.D.); 2Department of Surgery, Albert Schweitzer Hospital, 3318 AT Dordrecht, The Netherlands; p.w.plaisier@asz.nl; 3Department of Pathology, Amsterdam University Medical Center, VUMC, University Amsterdam, 1081 HV Amsterdam, The Netherlands; d.naves@amsterdamumc.nl (D.D.N.); pc.kortman@amsterdamumc.nl (P.C.K.); mam.broeckaert@amsterdamumc.nl (M.A.M.B.); 4Department of Cardiothoracic Surgery, Amsterdam University Medical Center, VUMC, University Amsterdam, 1081 HV Amsterdam, The Netherlands; 5Department of Pulmonary Diseases, Amsterdam University Medical Center, VUMC, University Amsterdam, 1081 HV Amsterdam, The Netherlands; j.daniels@amsterdamumc.nl; 6Cancer Center Amsterdam, Amsterdam University Medical Center, VUMC, University Amsterdam, 1081 HV Amsterdam, The Netherlands

**Keywords:** pulmonary carcinoid, distant metastases, immunohistochemistry, prognosis, Ki-67

## Abstract

**Simple Summary:**

Evidence of prediction of disease recurrence after curative surgery in bronchial carcinoid is still limited. The aim of this study was to retrospectively investigate a set of markers as potential predictors of dissemination. This study confirmed that adding OTP, CD44, and Ki-67 to the carcinoid classification improved the identification of patients who are at risk for metastatic disease. Patients who did not develop metastasis in follow-up had typical carcinoids with proliferation index <5% and positive OTP and CD44. Atypical carcinoids with proliferation index ≥5% and loss of OTP and/or CD44 were at high risk for distant metastases. Such patients should be screened for metastatic disease at diagnosis and during follow up.

**Abstract:**

Background: Curatively treated bronchial carcinoid tumors have a relatively low metastatic potential. Gradation into typical (TC) and atypical carcinoid (AC) is limited in terms of prognostic value, resulting in yearly follow-up of all patients. We examined the additional prognostic value of novel immunohistochemical (IHC) markers to current gradation of carcinoids. Methods: A retrospective single-institution cohort study was performed on 171 patients with pathologically diagnosed bronchial carcinoid (median follow-up: 66 months). The risk of developing distant metastases based on histopathological characteristics (Ki-67, p16, Rb, OTP, CD44, and tumor diameter) was evaluated using multivariate regression analysis and the Kaplan–Meier method. Results: Of 171 patients, seven (4%) had disseminated disease at presentation, and 164 (96%) received curative-intent treatment with either endobronchial treatment (EBT) (*n* = 61, 36%) or surgery (*n* = 103, 60%). Among the 164 patients, 13 developed metastases at follow-up of 81 months (IQR 45–162). Univariate analysis showed that Ki-67, mitotic index, OTP, CD44, and tumor diameter were associated with development of distant metastases. Multivariate analysis showed that mitotic count, Ki-67, and OTP were independent risk factors for development of distant metastases. Using a 5% cutoff for Ki-67, Kaplan–Meier analysis showed that the risk of distant metastasis development was significantly associated with the number of risk predictors (AC, Ki-67 ≥ 5%, and loss of OTP or CD44) (*p* < 0.0001). Six out of seven patients (86%) with all three positive risk factors developed distant metastasis. Conclusions: Mitotic count, proliferation index, and OTP IHC were independent predictors of dissemination at follow-up. In addition to the widely used carcinoid classification, a comprehensive analysis of histopathological variables including Ki-67, OTP, and CD44 could assist in the determination of distant metastasis risks of bronchial carcinoids.

## 1. Introduction

Pulmonary carcinoids (PCs) are a heterogeneous group of well-differentiated neuroendocrine tumors, representing approximately 1–2% of all lung malignancies [1,2]. As opposed to high-grade large-cell neuroendocrine carcinoma (LC-NEC) and small-cell lung cancer (SCLC), bronchial carcinoid tumors behave in a relative nonaggressive and benign manner. According to the current World Health Organization (WHO) classification of thoracic tumors (fifth edition; 2021), atypical carcinoid (AC) can be distinguished from the low-grade typical carcinoid (TC) on the basis of histologic evaluation of mitotic count (TC 0–1 and AC ≥ 2 mitoses per 2 mm^2^) and presence of focal or punctate necrosis in resection specimens [3]. ACs are characterized by a higher metastatic rate and a relatively unfavorable prognosis compared to TCs, which is reflected by a lower 5 year survival rate for AC (76% vs. 94%) after curative intent resection. Although ACs carry worse prognosis and present with higher distant disease recurrence rates (range 1–6% vs. range 14–29%), the majority of ACs do not disseminate [4]. Although a recent publication of Chiappetta et al. showed a prognostic score for survival in lung carcinoids, the scientific evidence of the prognostic value of clinical–pathological features such as lymphatic involvement, tumor size, mitotic count, and accurate prediction of disease recurrence after curative surgery is still limited [4,5,6,7,8]. This results in annual follow-up for an extensive period of time. This limitation indicates the need for combined assessment of prognostic markers to further subdivide PCs into relevant clinical categories. While proliferative assessment based on Ki-67 immunohistochemistry (IHC) has emerged as a helpful tool for discriminating PCs from high-grade neuroendocrine carcinomas, there is, in contrast to gastroenteropancreatic neuroendocrine tumors, no consensus concerning its prognostic value in low-grade PCs [9,10,11,12]. On the basis of recent gene and expression profiling of PCs, novel markers such as orthopedia homeobox (OTP) and CD44 have been identified as potential predictors of adverse outcomes [13]. In addition, IHC of retinoblastoma protein (Rb) and p16 might be helpful in predicting adverse outcomes due to divergent aberrations in the Rb/p16/cyclin D1 pathway among low- and high-grade neuroendocrine tumors [14,15,16].

The aim of this study was to retrospectively investigate a set of markers as potential predictors of dissemination in our institutional cohort. In particular, we analyzed the value of morphological, morphometric (mitotic count), and immunohistochemical (OTP, CD44, Ki-67, Rb, and P16) markers as potential prognostic indicators.

## 2. Material and Methods

### 2.1. Study Design and Methods

After approval by the Institutional Review Board (Medical Ethics Review Committee of Amsterdam UMC IRB00002991), patients referred to our tertiary referral center for curative treatment of bronchial carcinoid between 1991 and 2019, with sufficient tissue for additional immunohistochemistry, were included. Patient selection and curative treatment were previously described [17]. After curative treatment, patients were followed up annually with CT scan and, if indicated, with bronchoscopy. For the current analysis, we divided this patient cohort into three groups: (1) curatively treated patients without metastases during follow-up, (2) curatively treated patients who developed metastases during follow-up, and (3) patients diagnosed with disseminated disease at baseline (stage IV).

### 2.2. Case Evaluation and Definitive Diagnosis

Cases were analyzed and diagnosed (TC/G1 vs. AC/G2) by two pathologists specialized in pulmonary pathology, blinded to patient outcome. For stage classification, the eighth edition of the American Joint Commission on Cancer TNM staging system for non-small-cell lung cancer was used.

### 2.3. Mitotic Count and Presence of Necrosis

Mitotic count was scored by a specialized pathology laboratory technician (M.A.M. Broeckaert) as previously reported [18]. In short, the whole slide was first explored for mitotic hotspots, and the mitotic count was subsequently performed in the hotspot. Presence of necrosis was assessed by one of the pathologists.

### 2.4. Immunohistochemistry

Clones and conditions for IHC staining of Ki-67, Rb, p16, OTP, and CD44 are presented in Appendix A. Immunohistochemical staining intensity was scored using the H-score (range 0–300). In short, the percentage of cells at each staining intensity level (0, 1+, 2+, and 3+) was scored, and the H-score was calculated using the following formula: 1 × (% cells 1+) + 2 × (% cells 2+) + 3 × (% cells 3+) [19]. A staining intensity of <30 was considered negative for markers with overall high expression (OTP, CD44, and Rb) [20]. For p16, any positive H-score was classified as positive, as it was largely negative in almost all patients. The Ki-67 scoring was based on an estimated percentage of positive cells in the hotspot region after scanning the whole slide. The highest recorded value was taken into account, as described before [21].

### 2.5. Statistical Analysis

The statistical analyses and calculations were performed with SPSS 26.0 (IBM Corp., Armonk, NY, USA). Chi-square and Fisher tests were applied for comparison of qualitative variables. The area under the receiver operating characteristic curve (ROC), abbreviated as AUC, was calculated to identify the discriminatory ability of immunohistochemical markers and mitotic index for development of distant metastases during follow-up. Spearman’s correlation coefficient was used to measure the monotonic association with respect to continuous variables (mitotic and proliferation indices). A binary multiple regression analysis was performed to identify independent risk factors for distant metastases. Survival analysis was performed using the Kaplan–Meier method and log-rank test. Patients with disseminated disease at presentation were excluded from the survival analysis (*n* = 7). A *p*-value < 0.05 denoted statistical significance.

## 3. Results

### 3.1. Patient Characteristics

A total of 171 patients with bronchial carcinoid were included in the present study (Table 1). The median age at diagnosis was 49 (IQR: 36–61 years), and 97 (56.7%) patients were female. TC was diagnosed in 112 (65%) and AC in 59 (35%) patients. Immunohistochemistry results of both TC and AC are shown in Appendix A. A total of 164 (96%) patients underwent curative intent treatment with either definitive endobronchial therapy (EBT) (*n* = 61, 36%) or surgery (*n* = 103, 60%). Of 164 curatively treated patients, 13 (8%) developed metastases during follow-up (median FU 81 months, IQR: 45–162). Seven patients (4%) were diagnosed with distant metastases at presentation (liver *n* = 2, pleura *n* = 2, dura mater *n* = 1, skin *n* = 1, and adrenal gland *n* = 1) (Table 1).

### 3.2. Mitotic Index and Ki-67

Tumors of patients who developed distant metastases during follow-up (*n* = 13) showed a higher mitotic rate (median 8; IQR 2–10) compared to patients without metastases at follow-up (median 0; IQR 0–2, *p <* 0.001) (Table 2). Development of metastases was also associated with an increased Ki-67 index (median 6%; IQR 1–11) compared to non-metastatic cases (median 1%; IQR 1–2) (*p* < 0.001). As a result of considerable overlap in frequency distributions of Ki-67, the median Ki-67 indices for TC (1%; IQR 1–3) and AC (2%; 1–4) were not significantly different (*p* = 0.134) (Appendix A). The ROC analyses demonstrated that mitotic count (AUC 0.87; *p* < 0.0001) and Ki-67 (AUC 0.77; *p* = 0.0012) were discriminatory for development of distant metastases (*n* = 13) (Figure 1A). The optimal cutoff values from this analysis were analyzed using the ROC curve. Mitotic count of ≥2 per 2 mm^2^ and Ki-67 index of ≥5% were significantly associated with the occurrence of distant metastases (*p* < 0.001). Notably, mitotic count and Ki-67 showed negligible correlation (ρ = 0.19, *p* = 0.01) (Appendix A).

### 3.3. Immunohistochemistry

Loss of OTP and CD44 expression was observed in 30 cases (18%) (Table 2). In contrast to curatively treated patients without metastases at follow-up, loss of OTP (77 vs. 12%) and CD44 (69% vs. 10%) expression was significantly more common in patients with distant metastases (*p* = 0.004 and *p* < 0.001 respectively). ROC analyses revealed similar area under the curve (AUC) values for OTP (AUC 0.78, *p* = 0.001) and CD44 (AUC 0.82, *p* < 0.001) when development of distant metastases during follow-up was taken as an endpoint (Figure 1B). Rb was positive in all cases (100%), and 51 cases (30%) cases were positive for p16 staining. Both Rb and p16 were not discriminant for tumor classification (TC vs. AC), and no association with metastases at follow-up was observed (Table 2 and Appendix A).

### 3.4. Risk Factors for Distant Metastases

Subsequently, multivariate binary logistic regression was employed to identify independent risk factors of distant metastasis development during follow-up (*n* = 13). Input variables were all first tested in a univariate fashion for association with occurrence of distant metastases. Univariate analysis demonstrated that tumor diameter (*p* < 0.001), CD44 (*p* < 0.001), OTP (*p* < 0.001), mitotic count (*p* < 0.001), and Ki-67 (*p* < 0.001) were significantly associated with distant metastases at follow-up (Table 2). Of these significant terms, only mitotic count (*p* = 0.001), OTP (*p* = 0.004), and Ki-67 (*p* = 0.034) were independently associated with a higher risk of distant metastases (Table 2). In comparison to individual markers, this combination of predictors demonstrated a greater performance in terms of discriminatory capacity (AUC 0.90; CI 0.78–1.00; *p* < 0.0001) (Figure 1C).

### 3.5. Immunohistochemistry Profile in Patients with Disseminated Disease at Baseline

Univariate analysis revealed that CD44 (*p* < 0.001), Ki-67 (*p* < 0.001), and tumor diameter (*p* = 0.026) were significantly associated with disseminated disease at baseline. These significant variables on univariate analyses became nonsignificant on multivariate analysis. When discriminatory markers were compared between patients who presented disseminated disease (*n* = 7) and patients who developed metastases during follow-up (*n* = 13), an increased median proliferation index was found for Ki-67 in both groups (10%, IQR 8–15 vs. 6%, IQR 1–11; *p* = 0.157) (Figure 2A). Loss of OTP and CD44 was comparable in both groups, despite a nonsignificant trend in more frequent loss of CD44 in patients with disseminated disease at diagnosis (Figure 2B).

### 3.6. Metastasis-Free Survival

Lastly, a survival analysis was performed to assess the influence of independent predictors (Ki-67, mitotic count, and OTP) on the development of distant metastases in curatively treated patients. Although not significant in the multivariate analysis, but highly discriminative in the AUC, CD44 was also added to the survival analysis (Figure 3). The log-rank test showed a significant difference in distant metastasis-free survival in biomarker-based categories of patients (*p* < 0.0001). Patients (*n* = 76) with carcinoids harboring a favorable marker profile (Ki-67 < 5%, mitotic count < 2 per 2 mm^2^, and OTP or CD44 positivity) exclusively did not develop distant metastases during follow-up. Presence of only unfavorable biomarkers (Ki-67 ≥ 5%, mitotic count ≥ 2 per 2 mm^2^, and OTP or CD44 negativity) was associated with development of distant metastases in six out of seven patients (86%). Patients with at least one of these unfavorable prognostic characteristics developed metastases in seven (six AC and one TC) out of 79 cases (9%). This patient category is further specified in Appendix A. Figure 4 and Figure 5 show examples of favorable and unfavorable profiles of mitotic count, Ki-67, OTP, and CD44. 

## 4. Discussion

In this study, we demonstrated that combining mitotic count with proliferation index (Ki-67) and novel immunohistochemistry for OTP and CD44 greatly aided the prediction of metastatic dissemination in patients with bronchial carcinoid. None of the TCs with a low Ki-67 expression (<5%) and OTP/CD44 positivity metastasized during follow-up. In contrast, patients who presented with disseminated disease at baseline or developed metastases during follow-up could be identified by atypical carcinoid morphology in combination with a high proliferation index (Ki-67 ≥ 5%) and loss of OTP and/or CD44 expression. 

In this series of 171 carcinoids, we could show that loss of OTP and CD44 expression was strongly associated with unfavorable disease outcome. Loss of OTP was independently associated with development of distant metastases during follow-up. While only OTP was in independent predictor of dissemination at follow-up in this cohort, CD44 showed a similar pattern of loss in all but one patient. Therefore, the two biomarkers most probably show comparable prognostic value, and future studies should further elucidate this matter. In agreement with our results, gene expression profiling of carcinoids by Swarts et al. revealed strong downregulation of OTP and CD44 in association with poor survival and increased risk of metastases, and subsequent validation using immunohistochemistry confirmed the prognostic value of these promising biomarkers [13,22]. Similarly, Granberg et al. described a decreased progression-free and overall survival in TCs with loss of CD44 [23]. In more recent study by Papaxionis and colleagues, CD44 and OTP were incorporated with carcinoid gradation in a multivariate model with similar results [22]. In this study, however, the prognostic value of these markers was enhanced by the addition of proliferative index. The exact mechanisms of the homeobox protein family member contributing to aggressive tumor behavior and the molecular interactions with CD44 remain to be elucidated. Of these novel markers, CD44 is easily applicable in the current clinical practice, because of the wide availability of monoclonal antibodies and the possibilities for standardization of the staining procedure.

Ki-67, together with mitotic count, is already part of a grading system for gastrointestinal neuroendocrine tumors [24]. While this nuclear protein plays an essential role in the control and timing of cell proliferation, Ki-67 was previously not included in the World Health Organization classification scheme of PCs, partly because of inconsistencies in proposed cutoff thresholds and prognostic values [9,11,25]. However, the recent fifth edition WHO guideline for thoracic tumors currently suggests that Ki-67 ≥ 5% is most probably an AC [3,26]. Here, we report an association between Ki-67 ≥ 5% and development of distant metastases. A similar Ki-67 index cutoff of 5% was described by Dermawan et al. to predict long-term recurrence in PCs [27]. According to several studies, this 5% Ki-67 cutoff might enable a more refined three-tier histologic grading of carcinoids, in which PCs are stratified into three grades on the basis of histologic grade and Ki-67 immunoreactivity [27,28]. In this carcinoid cohort, Ki-67 did not exceed 20%, in concordance with the suggested diagnostic cutoff value for differentiation between carcinoids and large-cell neuroendocrine carcinoma (LCNEC) [29]. Notably, there was no correlation between the mitotic count and proliferation index. A possible explanation for this might be that Ki-67 is expressed in multiple phases of cell division, while mitotic count reflects activity of the mitotic phase only [30]. This is probably the reason for the independent and, in fact, additive prognostic value of Ki-67 and mitotic count.

With respect to the Rb/p16/cyclin D1 pathway, abnormalities in the form of RB and p16 loss were not associated with development of metastases or differentiation between TC and AC. A few studies, however, reported loss of tumor suppressor RB in a significant subset of ACs, while RB demonstrated preserved function as a low-abundance nuclear transcription factor in TCs [15,31]. The p16 protein, which negatively regulates the cyclin D-dependent phosphorylation of Rb, was partially positive in approximately 30% of cases. Interestingly, p16 did not show strong diffuse immunoreactivity and, therefore, might function as an adjunctive tool in differentiating between PC and LCNEC, in which diffuse positivity of p16 is more frequently observed [16,31]. 

To place our results in clinical context, we should explore how these outcomes might change the management of patients with bronchial carcinoid. First, biomarkers could be of help in the selection of the most appropriate treatment modality. In the current study, no dissemination of carcinoids harboring a favorable biomarker profile was detected after EBT or surgery. Therefore, patients with local intraluminal bronchial carcinoid, no signs of lymph node involvement, and a favorable biomarker profile may be good candidates for EBT, while all other patients with local disease should be treated with oncological surgical resection. Of course, prospective trials are required to assess safety and efficacy of such a biomarker-driven treatment strategy. Second, this biomarker set could also change the follow-up after radical treatment. Because local recurrence occurs in a small proportion of patients after EBT, follow-up with bronchoscopy and/or CT is obviously mandatory. After radical surgical resection, however, extensive follow-up may be futile in case of a carcinoid with a favorable biomarker profile (low mitotic count, low Ki-67 expression, and normal OTP/CD44 expression), which was 47% in our cohort.

The findings of the present study must be interpreted in the context of several potential limitations. First, since the patients were referred to our hospital for treatment with EBT, the population included in this study was exposed to selection bias. Carcinoids treated with EBT were predominantly small TC tumors compared to the subset of malignancies that were subjected to surgical resection. However, this allowed us to investigate the prognostic value of morphological, morphometric, and immunohistochemical characteristics in the context of low T stage and long/intensive follow-up. Second, as previously described, accurate assessment of Ki-67 and mitotic count can be problematic due to interobserver variability [32]. Third, a limited number of distant metastases served as the sample size for the multivariate analyses, reducing the power of the multimodal biomarker and increasing the margin of error. Considering the indolent growth of these tumors and the median follow-up time of 5.5 years in this study, we encourage further analysis of long-term results in an independent study.

## 5. Conclusions

Adding OTP, CD44, and Ki-67 to the widely used TC/AC classification provides a multimodal biomarker strategy that improves prognostic classification. In the future, this strategy may enable risk stratification and, hence, guide a more tailored approach regarding treatment and follow-up of patients with bronchial carcinoid.

## Figures and Tables

**Figure 1 cancers-14-03234-f001:**
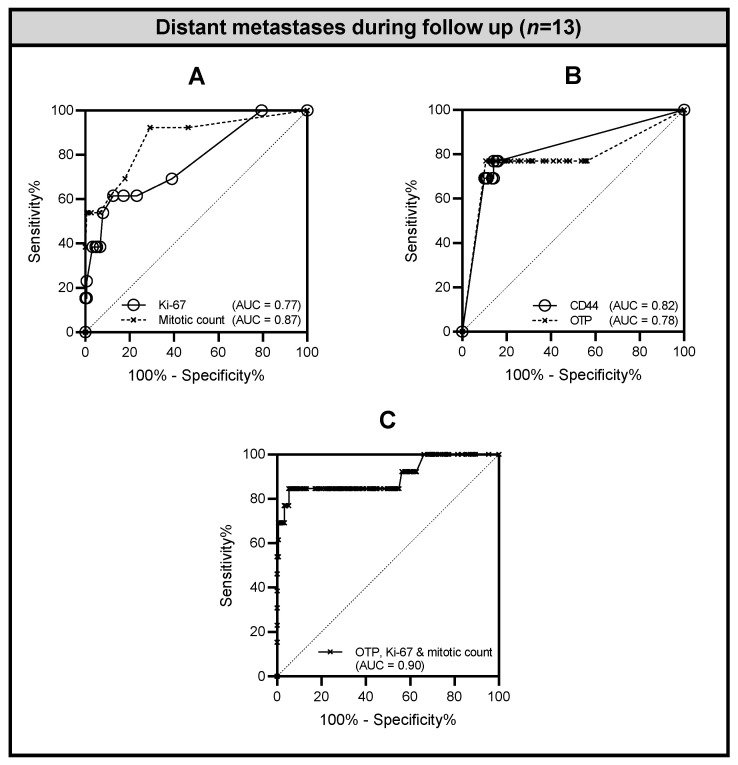
Receiver operating characteristic (ROC) curves for Ki-67, mitotic count, CD44, and OTP. Panels (**A**–**C**) present ROC curves of individual ((**A**) Ki-67, *p* = 0.003, and mitotic count, *p* < 0.000; (**B**) CD44, *p* < 0.000, and OTP, *p* < 0.000)) and combined markers ((**C**) OTP, Ki-67, and mitotic count, *p* < 0.000) for distinguishing occurrence of distant metastases (*n* = 13) from no occurrence of distant metastatic disease during follow-up.

**Figure 2 cancers-14-03234-f002:**
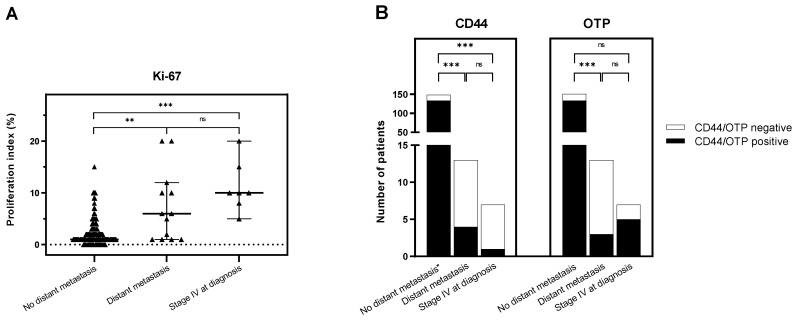
Immunohistochemical analysis of Ki-67 (**A**), and CD44 and OTP (**B**) expression in the studied patient categories; significance was determined using the Mann-Whitney U test (**A**) and Chi-Squared or Fisher’s Exact test (**B**); ns = not significant; ** *p* < 0.01; *** *p* < 0.001. ns: *p* > 0.05.

**Figure 3 cancers-14-03234-f003:**
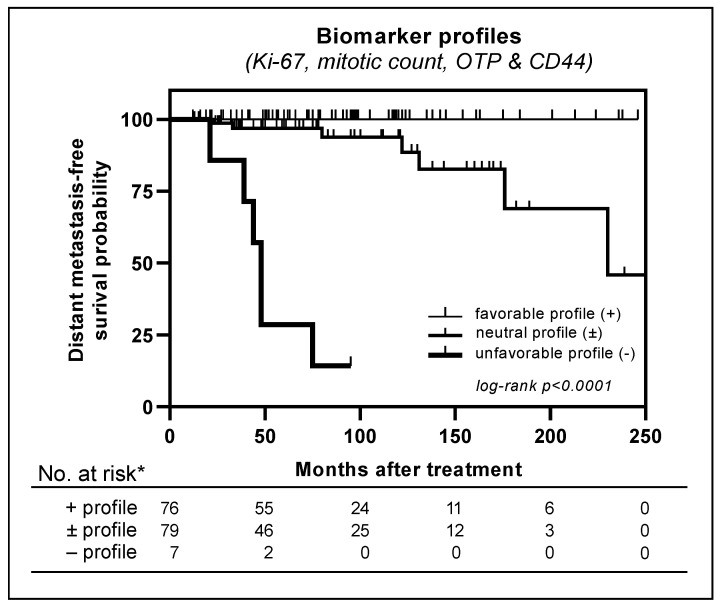
Distant metastasis-free survival curves estimated using Kaplan–Meier method for three biomarker profiles; + (favorable) profile: Ki-67 < 5%, mitotic count < 2 per 2 mm^2^, OTP and CD44 positivity;—(unfavorable) profile: Ki-67 ≥ 5%, mitotic count ≥ 2 per 2 mm^2^, and loss of OTP and/or CD44 expression; ± (neutral) profile: one or two characteristics of (un)favorable profile; * excluding nine patients due to stage IV disease at diagnosis (*n* = 7) or missing CD44 values (*n* = 2).

**Figure 4 cancers-14-03234-f004:**
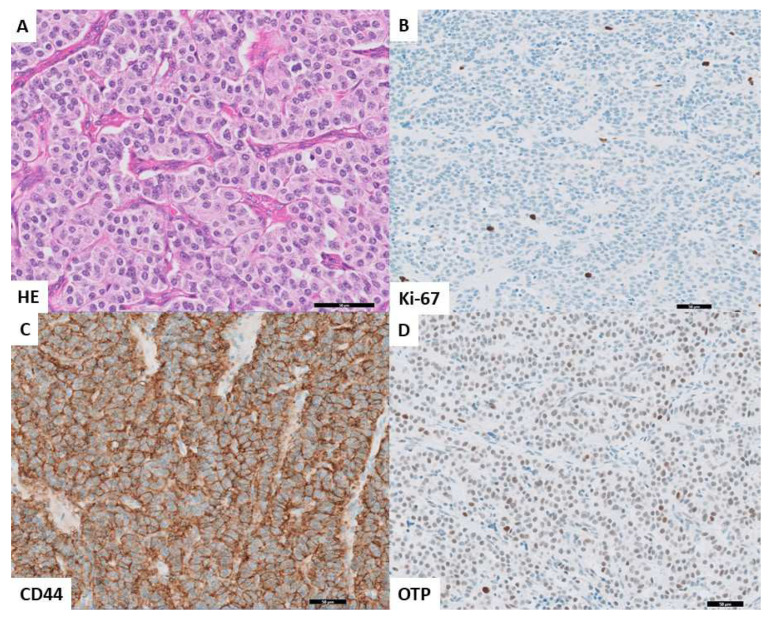
Prognostically favorable profile in histology and immunohistochemistry. (**A**). HE (40×) showing carcinoid with a well differentiated neuroendocrine morphology without mitoses. (**B**). Immunohistochemistry for Ki-67 (20×) showing a proliferation index of <3% in tumor cells. (**C**). Immunohistochemistry for CD44 (20×) with retained strong cytoplasmatic and membranous staining in the tumor cells. (**D**). Immunohistochemistry for OTP (20×) showing a retained nuclear staining in the tumor cells.

**Figure 5 cancers-14-03234-f005:**
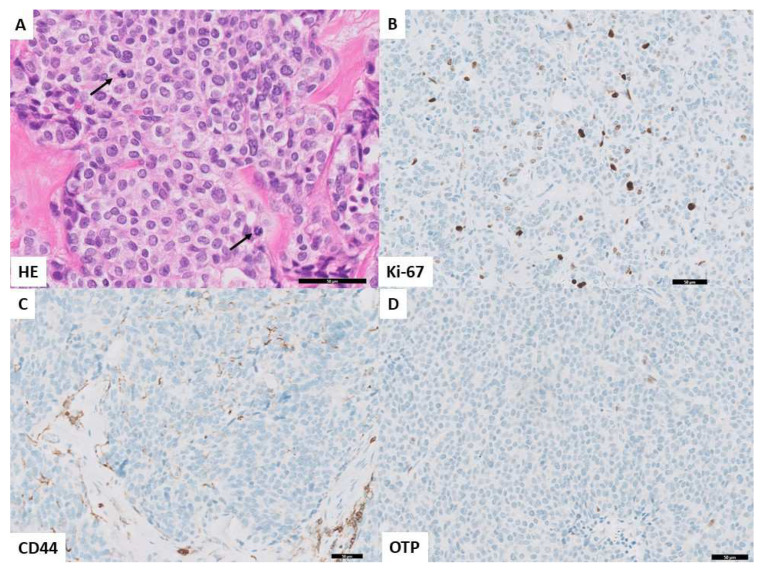
Unfavorable profile in histology and immunohistochemistry. (**A**). HE (40×) showing carcinoid with a mitotic hotspot with readily visible two mitotic figures in one HPF (arrows). (**B**). Immunohistochemistry for Ki-67 (20×) showing a proliferation index of >5% in tumor cells. (**C**). Immunohistochemistry for CD44 (20×) showing loss of cytoplasmatic and membranous staining in the tumor cells (**D**). Immunohistochemistry for OTP (20×) showing loss of nuclear staining in the tumor cells.

**Table 1 cancers-14-03234-t001:** Clinicopathological characteristics of patients treated for bronchial carcinoid (*n* = 171).

	Curative Treatment		Disseminated Disease after Curative-Intent Treatment		Distant Metastasis at Diagnosis		*p*-Value ^§^
**Number of patients (%)**	151	(88)	13	(8)	7	(4)	*NA*
Median age at diagnosis (IQR)	48	(35–60)	57	(44–67)	58	(47–64)	0.088
Gender (M/F)	67/84		4/9		3/4		0.618
**Histology**							<0.000
TC	107	(71)	1	(8)	4	(57)	-
AC	44	(29)	12	(92)	3	(43)	-
**Curative treatment**					*NA*		0.002
EBT	61	(40)	0	(0)	*-*		-
Surgery	90	(60)	13	(100)	*-*		-
**Type of surgery**					*NA*		0.190
Pneumonectomy	4	(3)	2	(15)	*-*		-
Bilobectomy	24	(16)	0	(0)	*-*		-
Sleeve lobectomy	18	(12)	3	(23)	*-*		-
Lobectomy	37	(25)	7	(54)	*-*		-
Bronchial sleeve resection	1	(1)	0	(0)	*-*		-
Bronchotomy	1	(1)	0	(0)	*-*		-
Segmentectomy	4	(3)	1	(8)	*-*		-
Wedge resection	1	(1)	0	(0)	*-*		-
**Tumor stage ^ⱡ^**							<0.000
1a	53	(35) *	0	(0)	0	(0)	-
1b	61	(40) *	6	(46)	0	(0)	-
1c	26	(17) *	4	(31)	0	(0)	-
2a	9	(6)	1	(8)	0	(0)	-
2b	1	(1)	0	(0)	0	(0)	-
3	1	(1)	2	(15)	0	(0)	-
4	0	(0)	0	(0)	7	(100)	-
**Median follow-up in months (IQR)**	66	(35–118)	81	(45–162)	50	(12–80)	0.268

EBT: endobronchial treatment; IQR: interquartile range; ^ⱡ^ TNM classification eighth edition; * In EBT cases, T and N status was based on tumor characteristics on CT scan; ^§^ calculated using the Kruskal–Wallis, chi-square, or Fisher’s exact test. *NA*: not applicable.

**Table 2 cancers-14-03234-t002:** Results of morphometric and immunohistochemical markers in different (prognostic) categories of bronchial carcinoid; significant *p*-values are reported in bold; curatively treated patients were selected as the reference group.

Variable	Patient Categories	IHC Positivity (%)		Median (SD)		Range (IQR)		Distant Metastases during FU (*n* = 13)	
								Univariate Analysis ^§^	Multivariate Analysis ^a^
Ki-67									
	Total cohort	*NA*		1	(3.8)	0–20	1–3		
	Curative treatment, no metastases at FU	*NA*		1	(2.5)	0–15	1–2		
	Distant metastases during FU	*NA*		6	(6.9)	1–20	1–11	<0.001	0.034
	Distant metastasis at baseline	*NA*		10	(4.9)	5–20	8–15		
Mitotic index									
	Total cohort	*NA*		1	(2.4)	0–10	0–2		
	Curative treatment, no metastases at FU	*NA*		0	(1.7)	0–8	0–2		
	Distant metastases during FU	*NA*		8	(3.9)	0–10	2–10	<0.001	0.001
	Distant metastasis at baseline	*NA*		1	(2.5)	0–7	0–2		
OTP									
	Total cohort	141/171	(82)	250	(112.9)	0–300	150–300		
	Curative treatment, no metastases at FU	133/151	(88)	260	(102.8)	0–300	160–300		
	Distant metastases during FU	3/13	(23)	0	(131.6)	0–300	0–150	<0.001	0.004
	Distant metastasis at baseline	5/7	(71)	270	(138.3)	0–300	0–300		
CD44									
	Total cohort *	139/169	(82)	300	(115.9)	0–300	300–300		
	Curative treatment, no metastases at FU	134/151	(89)	300	(92.6)	0–300	300–300		
	Distant metastases during FU	4/13	(31)	0	(138.7)	0–300	0–275	<0.001	*ns*
	Distant metastasis at baseline	1/7	(14)	0	(37.8)	0–100	0–0		
Rb									
	Total cohort	171/171	(100)	200	(79.1)	100–300	100–280		
	Curative treatment, no metastases at FU	151/151	(100)	190	(79.5)	100–300	100–280		
	Distant metastases during FU	13/13	(100)	190	(80.5)	100–300	100–265	0.893	*ns*
	Distant metastasis at baseline	7/7	(100)	240	(69.7)	100–300	190–300		
p16									
	Total cohort ^ⱡ^	51/170	(30)	0	(39.3)	0–240	0–5		
	Curative treatment, no metastases at FU	44/150	(29)	0	(34.5)	0–240	0–3		
	Distant metastases during FU	4/13	(31)	0	(74.9)	0–240	0–60	0.558	*ns*
	Distant metastasis at baseline	3/7	(43)	0	(36.5)	0–100	0–20		
Tumor diameter (mm)									
	Total cohort	*NA*		15	(11.1)	1–65	10–22		
	Curative treatment, no metastases at FU	*NA*		15	(9.7)	1–65	10–21		
	Distant metastases during FU	*NA*		27	(16.1)	14–65	15–35	<0.001	*ns*
	Distant metastasis at baseline	*NA*		25	(15.2)	11–55	16–36		

IHC: immunohistochemistry; SD: standard deviation; IQR: interquartile range; *NA*: not applicable; ns: not significant; * two missing CD44 values; ^ⱡ^ 1 missing p16 value; ^§^ calculated using the Mann–Whitney *U* test; ^a^ calculated using the binary multiple regression analysis (backward stepwise procedure).

## Data Availability

Data is contained within the article and the Appendix A.

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
