# Peer review of "A Multimodal Biomarker Predicts Dissemination of Bronchial Carcinoid"

_cancers, 2022, doi:10.3390/cancers14133234_

Round 1

Reviewer 1 Report

Dear Authors,

The study is interesting.

Here are my observations/suggestions/comments:

1.    Did you check the tumors in terms of TC/AC versus G1/G2 neuroendocrine tumors (the new terminology by WHO)? Was it an adequate grading as initial classification?

2.    Introduction

“Pulmonary carcinoids (PCs) are a heterogeneous group of well-differentiated neuro-endocrine neoplasms,”

Actually PC are well differentiated NET, as part of NEN classification, not well differentiated NEN

3.    Aim

“The aim of this study was to retrospectively investigate a set of markers as potential 89

predictors of dissemination in our institutional cohort. In particular, we analyzed the 90

value of morphological, morphometric (mitotic count) and novel immunohistochemical 91 (OTP, CD44, Ki-67, Rb and P16) markers as potential prognostic indicators.”

Ki67 is not a novel prognostic indicator; it is a traditional and well established prognostic marker (as single marker and/or in association with other markers)

4.    Do you think that your results surpass the limits of current WHO classification which does not take into account the value of  Ki67 for TC/AC?

5.    As a take home message, do you consider that each patient with TC/AC should routinely have immunohistochemistry analysis for OTP, CD44 in addition to Ki67?

Reviewer 2 Report

The comments are attached.

The authors tried to combine the mitotic index score and other IHC parameters to predict distant metastasis using a cohort study. The sample size of the FU is low.

However, there are some clarities required in this study. I would appreciate the authors answering the following questions.

1.       There are some typographical errors between 3.1 and table 1 with the median age and IQR at diagnosis please correct these.

2.       The distant metastasis during the FU was very less i.e 7/ 151 (~5%), although the case numbers look very high in the study, the target group has only a less proportion. Considering that the curatively treated bronchial carcinoids have low metastatic potential, how can this be relied except for the statistical significance shown.

3.       Since you have used IHC, for other predictors, was IHC used for Ki-67 as well in addition to the mitotic index? if not why was it skipped, how can you compare mitotic index with other IHC parameters.

4.       Could you explain about the IHC positivity rate as mentioned in table 2, was there a cut off for the positivity (low, medium or high).

5.       Ki-67 is a proliferation marker; therefore one would expect this in any proliferating cells, was Ki-67 measured at basal level and at followup level

6.       Authors did not mention if FU (metastasized samples) were retrieved to perform IHC, could you give clarity on this.

7.       How could you combine Ki-67 and OTB/CD44 as an indicator, do you have common value or scoring for these together

8.       Why are not there any IHC images shown here in the figures, it would be nice to represent some of the IHC images,

9.       The authors have used a 5% cut-off for Ki-67, What cut-off do you use for OTP an CD44?

Round 2

Reviewer 2 Report

The article can be accepted now.